# A Qualitative Preliminary Study on the Secondary Trauma Experiences of Individuals Participating in Search and Rescue Activities After an Earthquake

**DOI:** 10.3390/healthcare13101101

**Published:** 2025-05-09

**Authors:** Ebru Çorbacı, Ebru Tansel, Damla Alkan

**Affiliations:** 1Doctoral Program in Clinical Psychology, Cyprus Health and Social Sciences University, Güzelyurt 99750, Türkiye; 2Faculty of Social Sciences and Humanities, Cyprus Health and Social Sciences University, Güzelyurt 99750, Türkiye; ebru.tansel@kstu.edu.tr; 3Psychology Department, Faculty of Arts and Sciences, Final International University, Girne 99320, Türkiye; damla.alkan@final.edu.tr

**Keywords:** trauma, secondary trauma, post-traumatic growth, qualitative study

## Abstract

**Background**: This study aimed to analyze the challenges faced by professionals and volunteers in search and rescue operations after the earthquake that struck the southeastern region of Turkey, with its epicenter in Kahramanmaraş, on 6 February 2023. **Method**: This research was conducted using a qualitative approach; specifically, a phenomenological method. It presents the results of semi-structured individual interviews with eight sampled volunteers who participated in the search and rescue activities following the earthquake. Participants were between the ages of 24 and 45, and three were nurses, three were journalists, and two were civilian volunteers with no formal training in search and rescue. In terms of nationality, five participants were citizens of the Turkish Republic of Northern Cyprus (TRNC) only, while three had both TRNC and Turkish citizenship. Field duties included providing medical support, documenting incidents, and assisting survivors in collaboration with civil society organizations. The research data were analyzed using MAXQDA Analytic Pro 2020. **Results**: Within the scope of this research, four main themes and twenty-one sub-themes were identified. The first theme is related to the nature of the traumatic events and reflects the characteristics of the traumatic experiences of the participants. The second theme is secondary trauma symptoms, showing that the participants experienced symptoms such as overstimulation, intrusive thoughts, sleep problems, anger, and concentration difficulties. The third theme focuses on post-traumatic growth symptoms. Participants reported experiencing developmental changes following trauma, such as changes in self-perception, the ability to recognize new situations, understanding the value of life, and positive relationships related to personal growth. Finally, the fourth theme is related to the coping skills used to cope with traumatic events; participants shared their coping strategies and the impact of these strategies. **Conclusions**: This study highlights the need to assess individuals in search and rescue operations in terms of secondary trauma. Our findings may be used as a reference to develop post-disaster psychosocial support services for volunteer search and rescue teams. Additionally, the findings can be used to renew the content of pre-field preparation training.

## 1. Introduction

People who provide professional support services to individuals exposed to traumatic events (health workers [1], the police [2], mental health specialists [3], and social workers [4]) have a high risk of secondary traumatic stress symptoms among occupational groups.

The literature reflects that individuals working in disaster areas are negatively affected by spending time in crisis environments [5,6,7,8,9]. They are exposed to traumatic and negative life experiences at a higher rate than at other times when they leave their normal routines and work in these disaster areas [10]. Therefore, these individuals are at risk of experiencing secondary trauma symptoms. Secondary traumatic stress (STS) and its risk dimensions and protective factors in experiencing its symptoms have been revealed through comprehensive studies on responders after the September 11 attacks, which took place in human memory as the most resounding and destructive terrorist attack of the modern era [4,5,6,7,8,9,10,11]. In this context, inadequate supervision and social support, especially in individuals working in crisis and trauma areas, stand out as important factors that increase the level of secondary trauma [4].

Natural disasters such as earthquakes may lead to trauma effects, causing great material and moral losses. On 6 February 2023, earthquakes in 11 provinces in Turkey (Kahramanmaraş, Gaziantep, Şanlıurfa, Diyarbakır, Adana, Adıyaman, Malatya, Osmaniye, Hatay, Kilis, and Elazığ) caused many social, psychological, economic, and physical losses. As a result of these earthquakes, more than 44,000 people lost their lives, 2.2 million people were displaced, and 164,321 buildings were destroyed [12]. These figures are higher than the losses suffered in the 1939 Erzincan (Mw: 7.9) and 1999 Kocaeli (Mw: 7.6) earthquakes, which were the largest in Turkey in the previous century [13]. In the immediate aftermath, numerous state institutions and non-governmental organizations from across Turkey mobilized rapidly to provide aid and relief. The scope of the disaster prompted an international response, with several countries extending support and assistance. Among these was the Turkish Republic of Northern Cyprus (TRNC), whose involvement held unique emotional and symbolic significance.

What distinguishes the TRNC’s participation from that of other countries is the particular nature of its connection to the affected regions. Many individuals from the TRNC were personally impacted, either through the loss of relatives or close emotional ties to those affected. Additionally, the deep-rooted historical, cultural, and political relationship between Turkey and Northern Cyprus further intensified the emotional engagement of TRNC citizens. As a result, this might have led to empathic identification among volunteers from the TRNC who participated in the rescue work.

A tragic event that brought this emotional impact to a peak was the collapse of the Grand Isias Hotel in Adıyaman. The building’s destruction resulted in the deaths of 35 TRNC citizens, including members of the Gazimağusa Turkish Maarif College volleyball team who had traveled to Turkey to participate in a school sports competition [14]. This incident deeply affected the TRNC public, receiving extensive media coverage and generating a profound sense of collective grief [15]. In the days that followed, everyday life in the TRNC came to a halt; businesses closed, a period of national mourning was declared, and calls for justice for the victims became central to public discourse [16].

### 1.1. Trauma

According to the DSM-5, traumatic experiences may manifest in various forms, including direct exposure to life-threatening events involving death, serious injury, or sexual violence; witnessing such events; learning that they have occurred to close others; or repeated and intense indirect exposure to distressing details of the trauma [17,18].

### 1.2. Secondary Trauma

Secondary trauma is a psychological stress response that occurs in individuals who have not directly experienced a traumatic event themselves but are exposed to the traumatic experiences of others through close and empathic contact [19,20,21]. In the literature, professionals working in trauma-related helping professions who are indirectly and continuously exposed to traumatic content are at risk of developing symptoms of secondary traumatic stress [19,20,21,22]. Although the symptoms associated with primary and secondary trauma may overlap, there are important structural differences between the two. Figley (1995) distinguished between primary and secondary traumatic stress at the diagnostic level, noting that despite the similarity in symptomatology, such as re-experiencing, avoidance, and hyperarousal, the nature of exposure differs significantly [19].

For instance, while primary trauma involves direct exposure to a threat to one’s own life or physical integrity, secondary trauma arises through empathic engagement with the traumatic experiences of others, often resulting in similar psychological symptoms [4]. In both cases, individuals may experience intrusive thoughts, emotional numbing, and heightened arousal. However, in primary trauma, these symptoms originate from personal experience, whereas in secondary trauma, they result from indirect exposure to another’s trauma.

In this context, individuals involved in search and rescue operations are uniquely vulnerable as they may be exposed to trauma both directly by witnessing devastating scenes and indirectly through close contact with survivors and victims during rescue efforts [23,24,25,26,27].

### 1.3. Post-Traumatic Growth (PTG)

Human beings encounter different challenging events from birth and learn positive lessons. Thus, the positive impact of traumatic experiences has been an area of interest throughout history. Participation in search and rescue activities may lead to negative circumstances, although some individuals focus on growth and post-traumatic positive change rather than negative events [28]. The concept of post-traumatic growth focuses on positive cognitive, emotional, and interpersonal changes [29,30,31]. Another element is the importance of the traumatic event in negatively affecting the individual’s way of thinking regarding positive development in their life [30]. Post-traumatic growth can be analyzed under five different categories: self-positive perception, better interpersonal relationships, seeing new options, understanding the value of life, and questioning beliefs [28,29]. Studies regarding secondary trauma are mostly conducted by selecting one or two occupational groups [7,8,9,10], mainly aimed at understanding the psychological conditions of health workers [11,32].

From the perspective of this study, participants from different professional groups (press members and health workers) who voluntarily participated in search and rescue operations were covered. The research participants were also selected from those who joined search and rescue operations by traveling from the TRNC to Turkey. They were mobilized by the desire to help another country and, especially, the desire to rescue TRNC nationals. This characteristic puts this study in a different perspective. At the time of search and rescue activities, participants run the risk of becoming secondary victims of trauma. They not only have close contact with the victims but are also exposed to the direct negative effects of the event. Therefore, this study is a detailed examination of the traumatic experiences, challenging conditions, traumatic stress symptoms, post-traumatic growth, and coping styles of professionals and volunteers who participated in search and rescue operations during the Kahramanmaraş earthquake on 6 February 2023.

## 2. Materials and Methods

### 2.1. Research Design

This study was conducted using a qualitative method, with qualitative data collection methods, such as observation, interviews, or document analysis, utilized accordingly [33]. We used a qualitative research method known as a “phenomenological study” [34]. The phenomenological approach seeks to deeply understand individuals’ experiences, and this study specifically examines the challenges and traumatic effects faced by those involved in search and rescue activities. Through focusing on the differential impact of trauma and the emotional and psychological meaning-making of participants, this approach offers comprehensive insights into how secondary trauma is experienced.

### 2.2. Participants

As institutional participation in this research was not approved, only individuals who volunteered for search and rescue work with their own resources could be included in this study. Eight participants were reached using the snowball technique. This method is particularly effective for reaching hidden or difficult-to-access populations as it enables participant recruitment through a chain of referrals [35]. The participants arrived at the disaster area between 6 and 9 February and remained active in the field for periods ranging from 4 to 10 days. Nurses took on tasks such as providing basic medical services, administering first aid, and assisting in the care of the injured. Journalists were not only responsible for documenting the events and informing the public but also supported the processes of body bagging and victim identification, which involved emotionally and physically demanding tasks. Civil volunteers contributed to rubble removal, excavation support, and various humanitarian aid activities.

The criteria for inclusion were as follows:(1)Participation in search and rescue operations for at least 3 days after the earthquakes;(2)Active duty in the debris area;(3)Staying in the operation area for at least 5 days.

The participants’ demographic information (age, gender, occupation, and trauma history) is presented in Table 1.

### 2.3. Data Collection

The research process was explained to the participants in detail before the interviews, and their written consent was obtained. The data collection process was terminated when no new themes or information emerged during the interviews. During the analyses, the data were deemed both comprehensive and sufficient, and the data collection process was completed. Data were collected between 9 December 2023 and 28 December 2023 from eligible individuals who agreed to participate in this study.

The interviews were conducted in the first author’s private psychotherapy clinic, which provided a calm, confidential, and comfortable setting. Each interview lasted approximately fifty minutes and was conducted in a therapy room located within the clinic. As noted, data saturation was reached during this process, as recurring themes began to emerge and no new significant information was obtained from subsequent interviews.

Participants were explicitly informed that the interviews would be audio-recorded for research purposes only and that all data would be securely stored and kept confidential.

All interviews were conducted by the first author, a licensed clinical psychologist and certified positive psychotherapist. In addition to verbal content, the researcher took brief notes on participants’ emotional reactions and non-verbal cues throughout the sessions. When sensitive topics arose, the researcher maintained an empathic and supportive stance, pacing the conversation in accordance with the participant’s emotional state. If participants experienced emotional discomfort during the interviews, grounding or breathing techniques were applied as needed. Furthermore, participants were provided with information on how to access further psychological support if they desired. The researcher had access to a professional referral network, consisting of trusted therapists from various fields, and participants assessed as possibly needing further support were referred accordingly. No urgent psychological intervention was required during or following any of the interviews.

### 2.4. Data Evaluation

MAXQDA Analytical Pro 2020 was used to analyze the research data. Deductive and inductive methods were combined through descriptive analysis and content analysis. Moreover, themes were first determined using the descriptive analysis method. Then, the content analysis method was used for concepts and themes that could not be interpreted through descriptive analysis, as well as some unnoticed concepts and themes. The data were analyzed and explored more deeply with content analysis [33,34].

### 2.5. Validity and Reliability of Research Data

In qualitative research, various application steps ensure the reliability of data, such as credibility, reliability, confirmability, transferability, and transparency [36]. In this study, these steps were followed to ensure the reliability of the data. First, in line with the researchers’ professional experience, theoretical knowledge, academic study experience, and observations, we determined the physical, emotional, and social effects of participating in search and rescue activities on individuals’ lives and the impact of these effects on post-traumatic growth. To ensure the research’s reliability, the participants were included in the research according to the principle of voluntariness and were asked to detail their individual experiences in the process as much as possible.

Prior to the development of the interview questions, both the national and international literature were extensively reviewed. This process included an examination of measurement tools used in studies such as [19,22,28], as well as theoretical and thematic insights from research by [29,37,38]. The literature review guided the construction of this study’s theoretical framework and provided a comprehensive understanding of the research topic. Based on this framework, the interview questions were designed. To ensure the clarity, relevance, and content validity of the items, expert feedback was obtained from three field specialists. Necessary revisions were made in light of their evaluations, and the final version of the interview form was produced accordingly.

The semi-structured interview form that was prepared to explore the problems experienced during the subjects’ participation in search and rescue operations consisted of three main sections and 18 questions. In the first part, nine questions aimed to determine the sociodemographic characteristics of the participants and to identify whether they have previously intervened or witnessed traumatizing events (witnessing death, seeing severe injuries, dismembered faces and bodies, etc.) and whether they have received psychological support and first aid training before participating in search and rescue operations. In the second part, there were five questions about how they decided to participate in search and rescue operations; the region that they went to and the duration of their stay; whether they intervened or witnessed traumatizing events (witnessing death; severe injuries; dismembered faces; bodies; etc.) during the operations; what the challenging conditions were; and the traumatic events that affected the participants most deeply. In the third and last section, four questions were asked to determine how the traumatic events that the participants experienced affected their inner world, whether they received support (from professionals, family, friends, etc.) after the operations, how the events affected their private relationships, and how they coped with the events.

The researchers carried out the thematic analysis process independently of each other to increase the reliability of the research. After completing the analysis process independently, the researchers came together and finalized the findings. To ensure the validity and reliability of the study, four criteria suggested by Guba and Lincoln (1983) were considered [36]. Each researcher should analyze the data independently to ensure the reliability of the research data and the findings of the qualitative analysis [36]. After repeated readings, the researchers identified important expressions and coded them. During the coding phase, themes and their descriptions were generated. Finally, the researchers placed the participant statements under the themes and sub-themes. Each researcher carried out the steps up to this stage independently. After all these steps were completed, the researchers came together and compared the themes and sub-themes. In the final stages, comparing the data with previous stages, we examined whether new themes and codes were already present. By observing that no new information was added, we accepted that the data had reached saturation levels. During this process, we concluded that in reaching the data saturation level, new themes did not proliferate, and existing themes became more detailed. Data saturation allowed us to terminate this study at a point appropriate for our purpose. At this stage, we considered the diversity of emotional states and the in-depth analysis of the themes [33,34,35]. This study was conducted and reported in accordance with the COREQ (Consolidated Criteria for Reporting Qualitative Research) (see Appendix A), a 32-item checklist developed to enhance transparency and rigor in qualitative research reporting [39].

### 2.6. Ethical Compliance

This research was approved by the Scientific Ethical Committee on 8 December 2023 under file number (KTSU//2023/266). Following approval, all participants who planned to participate in this study were informed about this research, and their informed consent was obtained before the data collection phase, showing that they voluntarily participated in this research.

## 3. Results

This study included eight participants aged between 24 and 45. Three were nurses, three were journalists, and two volunteered in the disaster area. In response to the question “How did you decide to participate in search and rescue efforts?”, four participants stated that they took action with the sole intention of helping people in Turkey without knowing where they were going, three participants were in the region to fulfill their professional responsibilities, and one participant stated that he participated in search and rescue efforts to save his family. Within the scope of the study findings, the traumatic events experienced in the debris zone were primarily discussed in relation to the challenging conditions in that area. Findings on the nature of the traumatic events, secondary trauma effects, post-traumatic growth, and ways of coping with traumatic stress are also presented.

According to Table 2, all eight participants stated that cold weather and toilet problems were the most important obstacles among the challenging conditions in the debris field. Six stated that the debris zone resembled a war environment and that shelter posed a serious challenge. Five stated that aftershocks were a challenge, while four stated that not being able to use communication channels, not getting enough food, and the smell of dead bodies were challenges. Three stated that they were negatively affected by the lack of medical supplies and transportation problems.

In addition, two participants drew attention to the security problems and lack of coordination in the debris field and stated that conducting search and rescue operations in the same environment with earthquake victims was also among the challenging conditions.

According to Table 3, four themes (the nature of the traumatic events experienced, secondary trauma symptoms, post-traumatic growth symptoms, and coping with trauma) and twenty-one sub-themes were generated.

### 3.1. Characteristics of Trauma

Six sub-themes were obtained from answers to the question “Can you briefly describe the traumatic event that you experienced during your participation in search and rescue operations and that affected you most deeply?”. These themes are as follows: seeing a mutilated body, seeing dead children, seeing an orphaned lifeless body, the death of a familiar friend, unsuccessful rescue experiences, and the identification process of the relatives of earthquake victims (Figure 1).

#### 3.1.1. Seeing a Mutilated Body

Among the participants, four individuals stated that being exposed to a dismembered lifeless body was an event that deeply affected them.

“*Their faces were unrecognizable because concrete had fallen on their faces. A lot of wounds, or a crushed image. It is extremely bad*.”P4

Another participant expressed how seeing a dismembered human body negatively affected him mentally and psychologically:

“*The debris had smashed them all*.”P6

#### 3.1.2. Seeing Dead Bodies

Five participants stated that they were significantly affected by seeing too many dead bodies.

“*I looked up and there were dead bodies all along the pavements. It’s a screaming pandemonium. The pavements are full of corpses. The bodies were open so that anyone who could identify them could see them*.”P6

“*The bodies were on top of one another… “The bodies were maybe 10, 15 on top of one another. In front of every building. Ambulances would not be able to pick up the bodies. The corpses remain and of course they stink. People would come, naturally, and it was painful to see their reactions when they saw the corpses days later. I was very affected when I saw those people*.”P8

#### 3.1.3. Seeing Dead Children

Similarly, four participants noted that being exposed to lifeless children’s bodies affected them deeply.

“*There were 33 children. None of the children’s bodies were recovered whole. The debris smashed them all. X child was the one who affected me the most. The debris collapsed and rubble fell right in the middle of the bed, his head was severed, and he had six parts. Only the leg part came out. They scraped the top part of his head with a shovel. The pieces were put in a bucket or something*.”P6

“*There was a girl dressed in pink. She didn’t belong to our group. And I always wondered what the fate of that girl child was. The fact that we could not take care of it. The state of that girl. She has no mother, no father. Her lifeless body is lying there. It made me very sad that we couldn’t take care of her. I felt like we couldn’t take care of her. So, it stayed like that. They took her away. Who took her? We never knew where they took it. I think there must have been many similar incidents*.”P2

#### 3.1.4. Death of a Familiar Friend

Four participants stated that seeing their friends in the wreckage area affected them deeply.

“*Many of them were my friends. I saw all the deaths. I had a friend who was under the rubble with his child. When I found my friend, I didn’t recognize him. It was very strange, I found him on the fifth day, but I didn’t recognize him*.”P1

“*I knew most of the people there, which was very difficult. My friends, their children, my nephews. It is very difficult. Very*.”P7

#### 3.1.5. Unsuccessful Rescue Experiences

Five of the participants indicated that they were significantly affected by unsuccessful rescue experiences.

“*People from the next building. I heard people there. They shouted there for two days and died. Crying and shouting, but I can’t do anything. You can’t reach them. Everyone was like this*.”P1

“*The 12–13-story building collapsed in such a way that there were people in the basement, on the first and second floors. There is a shouting sound from below. 15–20 people shouting. They surrounded the place. They couldn’t save those people. It was very bad to watch it*.”P8

#### 3.1.6. Traumas of Relatives of Earthquake Victims

Two participants stated that seeing the helplessness of the relatives of the people affected by the earthquake affected them deeply.

“*I saw a man whose father was trapped by a column. He was diabetic. They fed him with a straw for 3 days. They gave him his medication. We heard him saying, “Save me, save me, save me, save me. He ate and drank for 3 days. But the man died. No one could do anything*.”P8

#### 3.1.7. The Identification Process of the Relatives of Earthquake Victims

All the participants stated that watching families identify their relatives was a traumatic event that deeply affected them.

“*A family looked at the face of their child. He said, “This looks like my son, let me see his t-shirt. His hair is like this. I didn’t recognize him, let me look at his hands. This is my son.” He started kissing his son on the wrapped three-day-old corpse*.”P3

“*They took the child out and started to say out loud what was on her. A 14-year-old girl has something red on her. He was saying that the family matching this description should come and identify her. But it was not like to be identified*.”P5

### 3.2. Findings on the Theme of Secondary Trauma Symptoms

Four sub-themes were generated based on answers to the research questions and the literature on the concept of secondary trauma, such as avoidance, arousal, reliving, and impairments in functionality. The sub-theme of impairments in functionality is divided into three sub-categories: sleep, anger, and concentration problems (Figure 2).

#### 3.2.1. Impaired Functionality

##### Sleep Problems

Five participants said that they had sleep problems after the search and rescue operations ended.

“*I have ongoing sleep problems from time to time. Not as often as before, but I still wake up from time to time*.”P4

“*I was sleepless for a while*.”P5

##### Concentration Problems

Two participants felt that they had concentration issues following the search and rescue activities.

“*I’m shattered because of lack of sleep. I couldn’t devote myself to my work, I couldn’t devote myself to my family and friends. I had a concentration disorder*.”P7

##### Anger

On the other hand, three participants mentioned that they had anger issues following their experiences.

“*I had an aggression problem for the first two months. I had an anger problem. I used to snap too much on unnecessary things. It passed in time. I was angry with people. People who knew me understood me. It made me a sharp, aggressive, and emotional person*.”P6

#### 3.2.2. Arousal

Four of the participants noted that they experienced signs of arousal after the search and rescue operations.

“*When I go somewhere, for example, I am here today, I started to think automatically if there is an earthquake at any moment, how can I get out of here, where is the door, where would it be better to get out, what floor am I on*.”P7

#### 3.2.3. Avoidance

Three participants indicated that they experienced symptoms of avoidance after the search and rescue operations.

“*I pretended it didn’t happen. I said you’ll have a healthier life if you shut it down. I felt like I would collapse if I got into it. That’s why I didn’t want to meet with the families or anything else. We never talked to my friend who went. He doesn’t bring it up and I don’t bring it up. That mission is behind us. Life goes on. And as soon as I arrived, I started to take care of many different things at work. I tried to distract myself with them. After I came back from there, I didn’t read a single news item about this issue. I didn’t contact them. I didn’t listen to their stories. I pretended they didn’t exist*.”P5

#### 3.2.4. Reliving the Moment

Three participants stated that they experienced signs of reliving their experiences even after the operations finished.

“*After the earthquake, especially in the first month, I got up a lot, jumping out of bed, wondering if an earthquake was happening? When I felt the slightest shaking, I wondered if an earthquake was happening. I still do?” I still haven’t deleted the photos. For example, I look at them from time to time. When I read anything about the earthquake, I look at those photos as they come to my mind. These photos remind me of what I told you. The beds, deaths, corpses, the state of the people where I was charging my phone reminds me of that mosque and many other things. Being there is not something easy to cope with*.”P4

### 3.3. Findings on the Theme of Post-Traumatic Growth

Four sub-themes were generated from answers to the question “How do the traumatic events you experienced in search and rescue operations affect your inner world today?” and the literature review related to the concept of post-traumatic growth. These sub-themes acknowledge the value of life, interpersonal relationships, the philosophy of life, and being able to see new options (Figure 3).

#### 3.3.1. Acknowledging the Value of Life

Seven participants stated that after being part of the search and rescue operation, they questioned the value of life and acknowledged it.

“*I realized that life could end at any moment, that it is only a moment, that it is not in your or anyone else’s hands. I realized that life is short, that it can end in a minute. I learnt not to be sad at the point of living life. I am grateful. I was aware of death, but I realized it more*.”P1

“*It reinforced my thoughts. Since I know that there will be death in life, I try to pursue whatever will give me happiness in my life. I am after spirituality; my profession has been a great binding at this point. I think even a study can be conducted on this*.”P2

#### 3.3.2. Improvement in Interpersonal Relations

Six participants mentioned that participation in the operation had positively impacted their lives.

“*It was a turning point for me. I decided to leave that day. The pain of some people was my turning point. Yes, I did not want such a thing to happen, but after it happened, I thought that I should move forward as much as I can help them and as much as I can help myself*.”P3

“*I started to be more sincere in my friendships. When you are frank and honest, you realize that you have nothing to lose. I don’t get into small-scale arguments anymore. Then I realized the value of family and friends more*.”P6

#### 3.3.3. Ability to See Self-Potential

Three participants concluded that their participation definitely made a positive contribution to seeing their own potential.

“*I grew up, I had no experience, now I have experience. It was a wild and horrible environment there. I allowed the events that happened to change me. I will never forget what I witnessed in Adıyaman throughout my life. I think it had a very positive contribution. I realized that I should work in a better institution*.”P6

“*After seeing them, I realized that I am really stronger, and I have to go on with life*.”P5

#### 3.3.4. Questioning Belief Systems

One of the participants stated that the traumatic events caused him to question his belief system.

“*I also lost some of my beliefs. After digging up all those dead children. I questioned my religious beliefs. Why would a child die in his bed? What sin did he commit that he died in his bed? I questioned what sins young people have committed*.”P1 Nurse

#### 3.3.5. New Options

A participant concluded that exposure to traumatic events helped him to see new options in life.

“*With most of the photographs I received awards and I have been offered jobs. It was to improve myself. I would consider leaving the organization that I am currently working for because I am not satisfied with the team*.”P6

### 3.4. Behaviors for Coping with Traumatic Events

Based on answers to the question “How did you cope with what you experienced after the search and rescue operations?” and the literature review on coping with trauma, four sub-themes were obtained: experiencing emotions, allocating time for hobbies, isolation, and alcohol use (Figure 4).

#### 3.4.1. Hobbies

Four participants stated that they spent time with their hobbies to cope with the traumatic events they were exposed to during the search and rescue operation.

“*I brought music into my life. Music is one of the reasons to hold on to life. Seeing a place, being in different places, getting to know different cultures*.”P4

#### 3.4.2. Feeling Emotions

Four participants indicated that they embraced their feelings to cope with the traumatic events during the search and rescue operation.

“*I said OK, I turned my head and left. I left crying*.”P1

“*Then I said, “Feel your emotions, if you need tears in your eyes, let them flow. Then do your job*.”P2

#### 3.4.3. Social Isolation

Two participants stated that they stayed away from people to cope with the traumatic events during the search and rescue operation.

“*I didn’t see anyone for 15 days. I didn’t see my mum, dad, or son*.”P1

“*I tried to live within myself*.”P2

#### 3.4.4. Alcohol Use

One of the participants reflected that he started using alcohol after experiencing traumatic events during the search and rescue operation.

“*I’ve been drinking alcohol. Actually recently*.”P7

## 4. Discussion

This study analyzes the experiences of participants in the search and rescue operation process after the Kahramanmaraş earthquake through thematic and descriptive analysis. We focused on the participants’ reasons for engaging in search and rescue operations, the types of traumatizing events that they intervened in, the conditions forced upon them during the operation, how the traumatizing events that they experienced affected their inner worlds and relationships, and their coping mechanisms. As a result of our interviews, four overarching themes were identified. In this section, the study themes will be compared with those of other studies.

### 4.1. Challenging Conditions

We found that among the challenging conditions in the debris zone, cold weather conditions affected the participants extremely negatively. Studies conducted with individuals participating in search and rescue activities have emphasized that cold and unpredictable weather conditions negatively affect people working in disaster areas [40]. Our participants reflected that they could not meet even their basic physical needs, such as toilets, food, sleep, and accommodation. Considering similar studies, we believe that the lack of basic needs impacts the physical and psychological well-being of participants [38,41,42].

The participants emphasized the chaotic environment of the debris zone and the lack of needs such as shelter, nutrition, sleep, toilets, and bathrooms. Being constantly on alert and the chaotic environment of debris zones bring psychological and physiological burdens [40]. According to Slepski [41], the scarcity faced by emergency responders negatively affects them. When working with trauma, adequate breaks should be taken, and heavy workloads should be considered accordingly [43]. Predicting the problems that may occur during rescue operations in advance and receiving the necessary training in this regard are important issues when determining needs in disaster areas, both for earthquake victims and the teams that will help them [38]. Disaster areas need to be organized quickly. At the same time, working in a disaster area involves being exposed to a heavy workload [44]. Moreover, participating in search and rescue activities in a different region also affects participants negatively. A study regarding the experiences of nurses working in the debris zone of Hurricane Sandy on 24 July 2012 revealed that helping patients in an unfamiliar working environment negatively affected the professional skills of the personnel [45]. To respond to a crisis in a healthy way in the disaster area, it is vital that individuals working in the field operate in a coordinated manner, as there is often a race against time [46]. The smallest mistake can have significant negative consequences. In 2019, a study on the problems experienced by paramedics while carrying out their treatment duties during disasters concluded that healthcare workers are largely at individual risk when making decisions about treatment tasks [32]. In this study, when the participants were asked which conditions were the most challenging, lack of coordination was identified. In the literature, multiple studies report that a lack of coordination negatively affects disaster area workers [38,47,48,49].

In this study, the participants were negatively affected by a lack of appropriate communication channels, which prevented them from communicating with their families and reporting their health status. This caused them to be unable to intervene if false news was spread about the debris field; thus, these factors had negative psychological effects on them. The literature shows that limited communication channels after an earthquake negatively affect work in the debris zone [1,38,47,50]. A study conducted using a qualitative research method on 11 people who participated in search and rescue activities after an earthquake that occurred in Nepal in 2015 emphasized that the inability of the media to convey precise and clear information without learning about the events in detail negatively affected aid workers and other individuals [40]. Notably, the unrealistic reporting of events negatively affects individuals in disaster areas and other individuals interested in this sub-topic [50].

We found that the presence of abandoned dead bodies in the surrounding areas negatively affected the study participants both in terms of their appearance and the unbearable smell produced. Similar results have been observed in other studies, where challenging conditions in a disaster area were investigated. Researchers emphasize that burial procedures should be carried out as soon as possible [5,32,38,47].

Another challenging condition experienced by the participants was a lack of medication and supplies. A study on the difficulties experienced in the double earthquakes that occurred in East Azerbaijan and Iran reflected the problems caused by medicine shortages and unsafe buildings during search and rescue operations [38].

This study found that working in areas where victims and their families were present was perceived as challenging by the participants and contributed to increased stress factors. A study conducted during a forest fire reflected that the presence of people in the event zone negatively affected work owing to a lack of knowledge and too many people being around [51].

### 4.2. Traumatic Events

According to our participants, seeing mutilated dead bodies was the most traumatic event during the rescue activities, mirroring results in the relevant literature [51]. Another major event is related to the exposure of children to trauma. Many studies show that child deaths, traumatic events to which children are exposed, and the helplessness experienced by children negatively affect participants [36,38,40,41,42,43,44,45,46,47]. Even encountering objects that used to belong to children in the area where the traumatic event occurred can negatively affect people [52]. This study also found that participants are deeply affected by seeing familiar faces or friends. A study performed with nurses who participated in earthquake rescue activities concluded that participants felt they needed to know that their families were well, which put them under pressure [47]. An inability to help people is another circumstance that deeply affected the participants. Considering the literature, we believe that witnessing the pain experienced by family members who lost their loved ones constitutes a risk factor in terms of secondary trauma [44,53].

### 4.3. Secondary Trauma Symptoms

We found that our participants exhibited prominent symptoms of secondary trauma, specifically issues such as sleep disturbances, difficulties with attention and concentration, anger management issues, hyperarousal, avoidance behaviors, and re-experiencing events. These symptoms are commonly recognized as classic indicators of secondary trauma, often observed in individuals who, while not directly exposed to trauma, are in close contact with those who have experienced it. Numerous studies on the concept of secondary trauma suggest that further research is needed regarding the diverse types of individuals affected [52,54,55]. In 2020, the secondary trauma and post-traumatic growth experiences of 13 search and rescue team workers who participated in disaster relief activities after an earthquake in Izmir were examined through in-depth interviews. As in this study, they found that the participants showed secondary traumatic stress symptoms, such as hyperarousal and out-of-control images and thoughts [56]. Our data align with recent studies conducted on individuals involved in search and rescue operations during the same earthquake [24,25,26,27,28,29,30,31,32,33,34,35,36,37,38,39,40,41,42,43,44,45,46,47,48,49,50,51,52,53,54,55,56,57,58]. These studies also reported elevated levels of secondary traumatic stress symptoms among professionals who, although not directly exposed to the traumatic event, were in close contact with affected individuals. The consistency of these findings underscores the importance of addressing secondary trauma not only at the individual level but also within organizational and systemic frameworks. Ensuring the psychological well-being of disaster response personnel emerges as a critical component in maintaining effective intervention capacities during crises.

### 4.4. Post-Traumatic Growth

The concept of post-traumatic growth includes positive changes in the life of an individual after trauma [42]. This study also revealed indications of post-traumatic growth among participants. In particular, the following dimensions come to the fore: understanding the value of life, questioning belief systems, changes in self-perception, realizing new possibilities, and improved interpersonal relationships. In a qualitative study conducted on individuals involved in search and rescue operations, it was revealed that they experienced post-traumatic growth in areas such as discovering personal power, understanding the value of life, and positive changes in interpersonal relationships, which are also among the categories of post-traumatic growth [42]. In a study conducted on individuals participating in professional search and rescue operations in Turkey, Yılmaz reported that the participants’ positive feelings about the traumatic event had a positive effect on post-traumatic growth [9]. A 2009 study on the Wenchuan earthquake found that post-traumatic growth symptoms gained after the earthquake reduced the effects of post-traumatic stress and that growth had an adaptive role in traumatic situations [59]. We also draw attention to other quantitative studies on post-traumatic growth that have demonstrated similar categories to those in this research [9,60,61].

### 4.5. Coping with Traumatic Events

When ways of coping with traumatic events are evaluated, the literature frequently emphasizes that trauma-based training is an important protective factor in coping with secondary trauma [1,3,52]. However, the participants in this study only received general emergency training and did not have any special training specific to disaster areas; this may have limited their capacity to cope with the negative effects of trauma exposure.

We obtained findings on our participants’ coping strategies regarding traumatic events. In particular, embracing feelings, turning to hobbies, alcohol use, and social isolation were prominent coping strategies. The ways of coping with the negative events experienced during search and rescue operations may be different for everyone. A study that investigated coping strategies with trauma revealed that individuals used exercises such as swimming, walking, and football to cope with trauma [46]. Researchers have drawn attention to the similarity of the strategies used by their participants to the strategies employed by primary victims [3,22,46]. In his research on secondary trauma, Hatipoğlu draws attention to the positive effect of physical activity in coping with secondary trauma. Participants noted that fully embracing their feelings helped them cope with traumatic events; thus, the literature emphasizes the awareness and acceptance of feelings [62]. A study conducted on emergency workers revealed that dysfunctional coping strategies negatively affect individuals [9]. For example, alcohol use leads to negative outcomes [62]; hence, it also stands out within the framework of this study as an unhealthy coping strategy. An increase in alcohol and drug use was observed in another study of secondary trauma [6]. Here, it is important to use healthy coping mechanisms to protect psychological health, and this study’s findings are consistent with the literature [55,56,58,59,60,61,62,63].

### 4.6. Strengths and Limitations

The litigation process initiated in Turkey at the time of this research, following the earthquakes at the Isias Hotel in Adıyaman, affected participation in search and rescue operations from Northern Cyprus. Thus, the institutional participation of the Civil Defense Organization and the Security Forces was not approved, and only those who agreed to participate individually were included in this research. This led to the exclusion of the experiences of public officials and limited our findings to professionals working independently of these organizations. This study is limited to only eight participants, hindering its generalizability. Additionally, all participants were male, and this study was conducted with only professionals and volunteers who traveled from Cyprus to Turkey, so further studies conducted with different demographic groups and geographies may yield different results. Nevertheless, this study provides valuable insights into post-disaster crisis management and contributes to the literature focusing on the experiences of professionals and volunteers involved in search and rescue operations after major disasters. In-depth interviews provided an important advantage in understanding the personal experiences of the participants, ensuring a wide range of data for this study. The diverse backgrounds of the participants allowed our findings to be evaluated from a broad perspective. The findings from the experiences of professionals and volunteers involved in the post-disaster process provide useful information for other groups and disaster management authorities who will face similar situations. Finally, the researchers’ careful selection of the sample and cautious approach to the participants ensured reliable results during the data collection process.

## 5. Conclusions

In 2023, two major earthquakes in Kahramanmaraş Pazarcık and Elbistan painfully reminded humanity that great material and moral losses can occur within a short period of time. This study analyzed the experiences of professionals and volunteers involved in search and rescue operations in detail. First, the traumatic events and challenging conditions faced by the participants were explored, as understanding such conditions is critical for search and rescue teams to act more quickly and effectively. Then, we evaluated the impact of these traumatic situations on the participants in terms of secondary trauma and post-traumatic growth symptoms, and information was also obtained regarding how the participants coped with their traumatic situations. This study is expected to significantly assist managers in the selection, training, and deployment of rescue teams to disaster areas. Our findings support the view that understanding traumatic situations will strengthen the preparedness of disaster responders and allow them to develop a more effective approach to interventions. The researchers suggest that future studies should focus more on the challenging conditions of disaster areas, the characteristics of traumatic events, and techniques for coping with these situations. Consequently, this study concludes that it is of great importance to provide psychological support and counseling services to all search and rescue teams. Participants should be trained about the difficulties that they may face before, during, and after the disaster and should be informed about stress management techniques and safety measures. Additionally, the physical and psychological difficulties faced by individuals who voluntarily participate in disaster areas without being affiliated with an institutional structure were an important focus of this study. The ways that the volunteers overcome difficulties are vital in the effectiveness of the search and rescue process and strategies to protect the health of volunteers. Accordingly, a special psychoeducation program for volunteers is recommended, including modules such as trauma, the symptoms of post-traumatic stress disorder, stress management, and post-traumatic resilience. The benefits of these sessions should be explained to volunteers and followed up on after training. In particular, stress management techniques such as mindfulness should be provided, and participants should be encouraged to use these techniques effectively.

## Figures and Tables

**Figure 1 healthcare-13-01101-f001:**
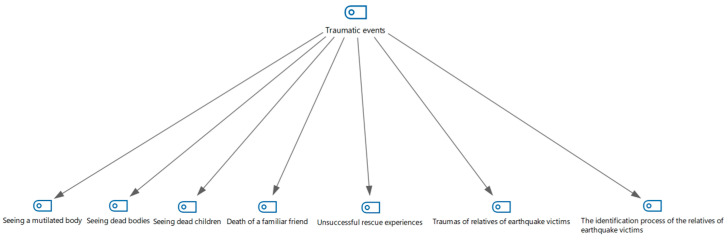
Traumatic events as shown in a hierarchical code–subcode model.

**Figure 2 healthcare-13-01101-f002:**
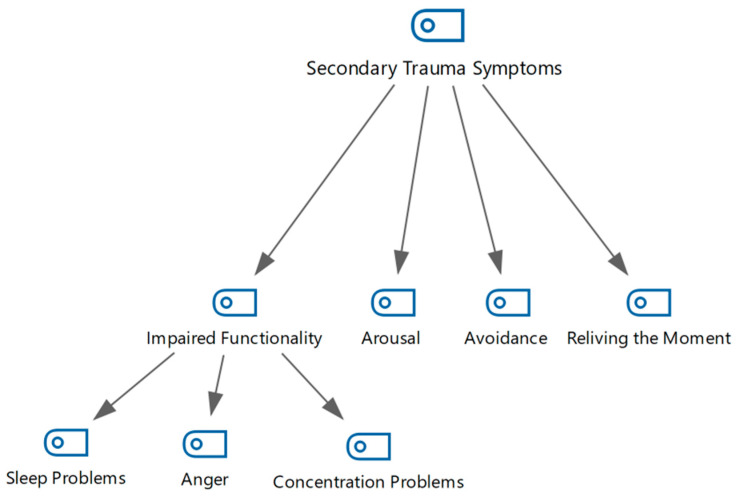
Secondary trauma symptoms as shown in a hierarchical code–subcode model.

**Figure 3 healthcare-13-01101-f003:**
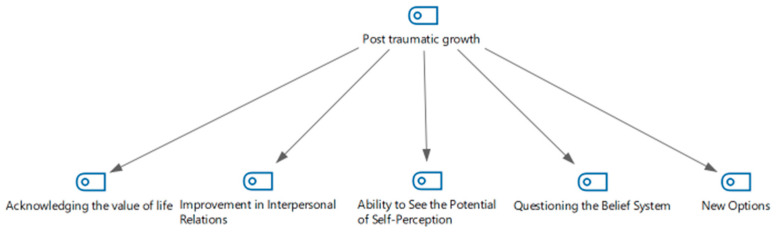
Post-traumatic growth hierarchical code–subcode model.

**Figure 4 healthcare-13-01101-f004:**
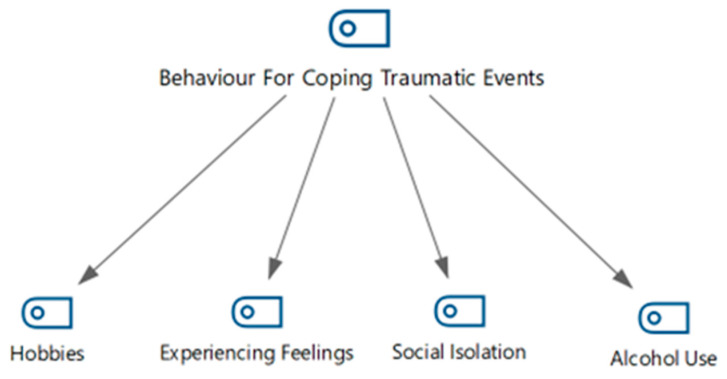
Behaviors for coping with traumatic events: hierarchical code–subcode model.

**Table 1 healthcare-13-01101-t001:** Descriptive characteristics of the participants.

Participant No.	Age	Marital Status	Nationality	Occupation	Rescue Operation Start Date	Working Time (Days)
P1	42	Widow	TRNC *	Nurse	6 February 2023	7
P2	37	Single	TRNC	Nurse	7 February 2023	6
P3	38	Married	TRNC	Nurse	7 February 2023	6
P4	32	Single	TRNC and Turkish	Journalist	7 February 2023	7
P5	32	Married	TRNC and Turkish	Journalist	6 February 2023	8
P6	24	Single	TRNC	Journalist	7 February 2023	6
P7	45	Married	TRNC	Volunteer	7 February 2023	4
P8	30	Married	TRNC and Turkish	Volunteer	9 February 2023	10

* TRNC: Turkish Republic of Northern Cyprus.

**Table 2 healthcare-13-01101-t002:** Document-based frequency table for challenging debris zone conditions.

Conditions	N	Frequency
Cold	8	10,000
Toilet	8	10,000
Chaos	6	7500
Accommodation	6	7500
Aftershocks	5	6250
Limited communication	4	5000
Dead body smell	4	5000
Food	4	5000
Lack of medication and supplies	3	3750
Transportation	3	3750
Security (extortion, despoiling)	2	2500
Lack of coordination at the excavation zone	2	2500
Search activity in the same environment with earthquake victims	2	2500
Analyzed documents	8	10,000

**Table 3 healthcare-13-01101-t003:** Themes and sub-themes.

Theme	Sub-Theme
Traumatic events	Seeing a mutilated body
	Seeing dead children
	Seeing the death of a familiar friend
	Not being able to help those who need help
	Traumas of relatives of earthquake victims/survivors’ experiences
	Process of families identifying their relatives
Symptoms of secondary trauma	Impaired functionality
	Sleep problems
	Concentration problems
	Anger problems
	Arousal
	Avoidance
	Living again
Post-traumatic growth	Understanding the value of life
	Questioning belief systems
	Self-perception
	Ability to see new options
	Improvement in interpersonal relationships
Coping with trauma	Embracing feelings
	Hobbies
	Alcohol use
	Social isolation

## Data Availability

The data supporting the findings of this study are available upon reasonable request. Researchers interested in accessing the data can contact Ebru Çorbacı at ebru_corbaci@hotmail.com. We are committed to promoting transparency and facilitating data sharing to further scientific investigation.

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
