# Peer review of "A Qualitative Preliminary Study on the Secondary Trauma Experiences of Individuals Participating in Search and Rescue Activities After an Earthquake"

_healthcare, 2025, doi:10.3390/healthcare13101101_

Round 1
Reviewer 1 Report
Comments and Suggestions for Authors
Thank you for considering the experience of recovery workers relative to trauma/post trauma symptoms, growth, and coping, after the horrible earthquake in Turkey. It is extremely important to add to the study of such matters, and the more international voices that are added to the research conversation, the better. It would be fascinating to see if there are specific symptoms/concerns of those assisting after earthquake, as opposed to other disasters. Natural and human-made disasters could potentially result in different concerns as one is, in effect, an interpersonal trauma, while the other is a natural disaster. So, this is an important area of work and the eventual development of a validated instrument for measurement and treatment direction is a wonderful aim.
Since this paper is aiming for publication in an English language journal, the language needs to be addressed as there are certain instances in which I cannot quite understand the meaning due to English errors. Most of the time I can figure out the meaning despite errors, but there are a few places in the text that I could not make out the meaning. Another over-all critique is that there are elements that need clearer definition of concepts. I think you may want to look at your using "secondary trauma" as the concept to define what these first responders experienced, vs it actually being a "primary trauma." Also there are many "landmark" studies about trauma overall as well as in first responders, that you might want to be sure to cite for your arguments.
Below are some specific elements that could be addressed.
Line 11 - One example of language: This first line of the abstract is understandable, but the word analyses should be "analyze," and the sentence, overall, could be made stronger and clearer. The first abstract sentence after "Method" needs a great deal of correction.
Line 16 - While this is not necessarily the place to do it, COREQ needs to be spelled out (the first time it appears) and explained.
Line 35 - It would be wonderful to have some citations right off the bat after that first sentence, as there is a great deal of literature on the post trauma sequalae of rescue/first responders. For one, there is a vast amount of data on the extensive rescue/response of 9/11 responders. It would be nice to set up what is known more globally, then to go to the specifics that are known relative to this particular disaster. Line 49 you bring in references about 2 other earthquakes, as well as this one. It would be great at the beginning, to use some of the broader published literature about rescue/recovery more broadly. Also, the references on line 49 - I would likely not include the doctoral dissertation. There is already a great deal of published literature that relates to this topic.
Line 55 - this paragraph could be more robust, defining trauma based on the icd-10 or the DSM 5.
Line 62 - a fundamental problem with this paper is relative to how you are defining secondary trauma. In fact, if you are going to define what these first responders experience as being secondary trauma, both primary and secondary trauma must be clearly defined. Based on current literature, the experience of first responders is usually a direct trauma. While they might also experience secondary trauma, they are directly exposed. According the dsm-5, a person may experience trauma if exposed to death, threatened death, actual or threatened physical injury, actual or threatened sexual violence when exposed either by direct exposure, witnessing the trauma, learning that the trauma has happened to a friend or loved one, or indirect exposure to aversive details, usually in the course of professional experience, like first responders. Generally, secondary trauma refers to what someone experiences if they only have that last criterion. Secondary trauma usually refers to the indirect experiencing of the suffering of others, for example as in a therapist who repeatedly listens to stories of trauma survivors. You could make an argument that it is secondary trauma, but if you are going to refer to this as secondary trauma, lay out the argument based on what you see in the literature.
88 - Can you define TRNC the first time you use it?
108 - can you definte the snowball technique?
lines 154-157 are repeats of the previous lines.
You include the questionnaire used in the appendix, but it would be great to have more about this questionnaire, and how you developed the question, etc, in the body of the text. Is it based on another qualitative questionnaire? On literature reviews?
413- in the discussion section, you mention "the reasons behind the decisions" to engage, etc, but this is the first mention of this anywhere. Again, please define the questions asked in the study. It is qualitative, so the whole thing is the content.
540 - the beginning of the strengths and limitations section - if this paper is for an international audience, please define the event you are referring to, or exlude it? The litigation that is mentioned.
A few more thoughts - the selection of the particular responders could be more fleshed out, including the differences between these kind of responders, which you do get at in the questionnaire, by asking about training of the responders, but again, this is buried in the appendix. Those trained in rescue/response are more immune to post trauma symptoms than those not trained in it, based on literature - it is a protective factor. Some of these folks might have been trained in disaster response, perhaps the nurses? Likely not the volunteers?
Comments on the Quality of English LanguageFor publication in English, the language must be improved. Perhaps a science interpreter could be employed for this purpose? Most of the errors are somewhat minor grammar or syntax errors, however, there are a number of errors that resulted in my not understanding the meaning of the text. It seems that an online translation service might have been used and unfortunately, there are times that, although the words might translate, the meaning does not.
Author Response
- Reviewer
General Commet
Since this paper is aiming for publication in an English language journal, the language needs to be addressed as there are certain instances in which I cannot quite understand the meaning due to English errors. Most of the time I can figure out the meaning despite errors, but there are a few places in the text that I could not make out the meaning. Another over-all critique is that there are elements that need clearer definition of concepts. I think you may want to look at your using "secondary trauma" as the concept to define what these first responders experienced, vs it actually being a "primary trauma." Also there are many "landmark" studies about trauma overall as well as in first responders, that you might want to be sure to cite for your arguments.
General Response
We thank the reviewer for this insightful comment regarding the conceptual clarity and the importance of referencing key foundational studies in the field.
In response, we have revised the manuscript to provide clearer definitions and distinctions between “primariy” and “secondary trauma.” Upon reflection, and considering the nature of the events experienced by the first responders in our study, we acknowledge that many of these experiences align more closely with “secondary trauma.” This distinction has now been clarified in both the theoretical framework and discussion sections.
Additionally, we have revisited the literature review to incorporate several landmark studies on trauma and its impact on first responders, including. These additions strengthen the theoretical foundation and support our key arguments with well-established research in the field.
Comment: Line 11 – One example of language: This first line of the abstract is understandable, but the word “analyses” should be “analyze,” and the sentence, overall, could be made stronger and clearer. The first abstract sentence after “Method” needs a great deal of correction.
Response: We thank our referee for pointing out the language issues in the abstract, particularly in the first sentence and the first sentence following the heading “Method.” We have carefully revised the abstract for clarity, grammar, and academic tone. The specific issue with the verb form (“analyses” vs. “analysis”) has been corrected, and the sentences in question have been restructured to improve readability and precision. In addition, the journal’s translation service was used.
Comment: The first abstract sentence after “Method” needs a great deal of correction.
Original sentence: "This study presents the results of semi-structured individual interviews with 8 purposively sampled volunteers that were participated to search and rescue activities."
Revised sentence: " This research was conducted using a qualitative approach; specifically, a phenomenological method. It presents the results of semi-structured individual interviews with eight sampled volunteers who participated in the search and rescue activities following the earthquake. Participants were between the ages of 24 and 45, of which 3 were nurses, 3 were journalists, and 2 were civilian volunteers with no formal training in search and rescue. In terms of nationality, five participants were citizens of the Turkish Republic of Northern Cyprus (TRNC) only, while three had both TRNC and Turkish citizenship. Field duties included providing medical support, documenting incidents, and assisting survivors in collaboration with civil society organizations. The research data were analyzed using MAXQDA Analytic Pro 2020. "
The change is found on page 1, line 15 of the article.
Comment: Line 16 - While this is not necessarily the place to do it, COREQ needs to be spelled out (the first time it appears) and explained.
Response: Thank you for pointing this out. We have removed the COREQ statement from the abstract. We introduced and explained this information at the beginning of the Method section.
Original sentence in the abstract: The study was conducted and reported using the COREQ checklist.
Revised sentence in the method: This study was reported in accordance with the COREQ (Consolidated Criteria for Reporting Qualitative Research), a 32-item checklist designed to ensure transparency and rigor in qualitative research reporting. The change is found on page 6, line 271 of the article.
Comment: Line 35 - It would be wonderful to have some citations right off the bat after that first sentence, as there is a great deal of literature on the post trauma sequalae of rescue/first responders.
Response: Thank you for your suggestion. Based on your suggestion, the relevant area has been expanded to highlight more globally known disasters. The relevant additions are on page 2, line 44 of the article.
Comment: Line 49 you bring in references about 2 other earthquakes, as well as this one. It would be great at the beginning, to use some of the broader published literature about rescue/recovery more broadly. Also, the references on line 49 - I would likely not include the doctoral dissertation. There is already a great deal of published literature that relates to this topic.
Response: Thank you for this helpful suggestion. In response, we have revised the related section by incorporating broader, peer-reviewed literature on rescue and recovery operations in disaster contexts, which provides a more comprehensive foundation for the study. Specifically, we have replaced the reference to the doctoral dissertation with more widely cited published sources in the field. These additions aim to better contextualize the study within the existing body of knowledge and enhance the academic rigor of the manuscript. The changes are on page [2], lines 47-57.
Comment: Line 55 - this paragraph could be more robust, defining trauma based on the icd-10 or the DSM 5.
Response: Thank you for your valuable contributions and constructive feedback aimed at enhancing the scientific depth of the study. In line with your comments, the relevant section has been restructured based on the diagnostic criteria of the DSM-5, and the necessary revisions have been implemented in the manuscript. Should further clarification be required, criteria from the ICD-10 are also ready to be incorporated in order to strengthen the conceptual framework of trauma.
Original paragraph: Trauma means the overall experiences that negatively affect the life of the individual in different ways and create shocking emotions such as fear, restlessness and anxiety [11,12,13]. At this point, the first question that comes to mind is whether every negative event experienced is trauma. Surely not [14]. In order for the negative event to be a trauma, it must be sudden, unexpected and uncontrollable as well as be of a nature that makes the person feel that he is under severe threat such as death or severe injury [13-15]
Revised paragraph: According to the DSM-5, traumatic experiences may manifest in various forms, including direct exposure to life-threatening events involving death, serious injury, or sexual violence; witnessing such events; learning that they have occurred to close others; or repeated and intense indirect exposure to distressing details of the trauma [16,17]. The changes are on page [2], lines 87-91.
Comment: Line 62 - A fundamental problem with this paper is relative to how you are defining secondary trauma. In fact, if you are going to define what these first responders experience as being secondary trauma, both primary and secondary trauma must be clearly defined. Based on current literature, the experience of first responders is usually a direct trauma. While they might also experience secondary trauma, they are directly exposed. According the dsm-5, a person may experience trauma if exposed to death, threatened death, actual or threatened physical injury, actual or threatened sexual violence when exposed either by direct exposure, witnessing the trauma, learning that the trauma has happened to a friend or loved one, or indirect exposure to aversive details, usually in the course of professional experience, like first responders. Generally, secondary trauma refers to what someone experiences if they only have that last criterion. Secondary trauma usually refers to the indirect experiencing of the suffering of others, for example as in a therapist who repeatedly listens to stories of trauma survivors. You could make an argument that it is secondary trauma, but if you are going to refer to this as secondary trauma, lay out the argument based on what you see in the literature.
Response: Thank you very much for your feedback. We really appreciate your contribution regarding the categorization of traumatic experiences among first responders. We specifically chose to use the concept of secondary trauma in this study because participants reported indirect exposure to traumatic events through their contact with victims rather than direct exposure. However, to ensure terminological clarity, we have carefully revised the relevant sections to provide a clearer explanation.
Revised Paragraph:
Secondary trauma is a psychological stress response that occurs in individuals who have not directly experienced a traumatic event themselves, but are exposed to the traumatic experiences of others through close and empathic contact [19,20,21]. In the literature, professionals working in trauma-related helping professions who are indirectly and continuously exposed to traumatic content are at risk for developing symptoms of secondary traumatic stress [19–22]. Although the symptoms associated with primary and secondary trauma may overlap, there are important structural differences between the two. Figley (1995) distinguished between primary and secondary traumatic stress at the diagnostic level, noting that despite the similarity in symptomatology—such as re-experiencing, avoidance, and hyperarousal—the nature of exposure differs significantly [19].
For instance, while primary trauma involves direct exposure to a threat to one's own life or physical integrity, secondary trauma arises through empathic engagement with the traumatic experiences of others, often resulting in similar psychological symptoms [4]. In both cases, individuals may experience intrusive thoughts, emotional numbing, and heightened arousal. However, in primary trauma, these symptoms originate from personal experience whereas, in secondary trauma, they result from indirect exposure to another's trauma.
In this context, individuals involved in search and rescue operations are uniquely vulnerable, as they may be exposed to trauma both directly by witnessing devastating scenes and indirectly through close contact with survivors and victims during rescue efforts [23,24,25,26,27]. The changes are on page [3], lines 94-114.
In this direction, the conceptual explanations of the article have been reconsidered starting from line 88; the concepts of trauma and secondary trauma have been clarified in the light of DSM-5 (APA, 2013) and current literature and added to the text.
Comment: Can you define TRNC the first time you use it?
Response: Thank you for your constructive suggestion regarding the clarification of the abbreviation 'TRNC' upon its first use. The relevant section has been revised accordingly to ensure clarity. The relevant abbreviation has been arranged as requested in the article where it was first written. The relevant edit was made on page 1, line 20 of the study.
Comment: Can you define the snowball technique?
Response: Thank you for your valuable feedback regarding the definition of the snowball sampling technique. In response, the sampling method used in the study has been clarified in the text, and the necessary revisions have been made accordingly.
Original paragraph: Eight people who went voluntarily to search and rescue operations with their own means were reached with the snowball technique, since the state institutions could not obtain a search warrant to reach the people who went to search and rescue operations
Revised Sentence:
Eight participants were reached using the snow-ball technique. This method is particularly effective for reaching hidden or difficult-to-access populations, as it enables participant recruitment through a chain of referrals [35]. The relevant edit was made on page 4, line 156-158 of the study.
Comment: lines 154-157 are repeats of the previous lines.
Response: Thank you very much for your careful reading and insightful comment.”- Lines 154-157 repeat information from previous lines”. We sincerely apologize for this oversight. It was an unintentional repetition resulting from the excitement and intensity of the writing process. The repeated lines have now been removed in the revised version of the manuscript.
Original paragraph: In order to ensure the validity and reliability of the study, four criteria suggested by Guba and Lincoln (1983) were taken as basis. It is recommended that each researcher should analyze the data independently of each other for the reliability of the research data and findings in the analysis of qualitative data [28]. Hence, two researchers analyzed the research data independently.
Revised Sentence:
To ensure the validity and reliability of the study, four criteria suggested by Guba and Lincoln (1983) were considered. Each researcher should analyze the data independently to ensure the reliability of the research data and findings in qualitative analysis [40].
The relevant edit was made on page 6, line 256-259 of the study.
Comment: You include the questionnaire used in the appendix, but it would be great to have more about this questionnaire, and how you developed the question, etc, in the body of the text. Is it based on another qualitative questionnaire? On literature reviews?
Response: I would like to sincerely thank the reviewer for this helpful comment. Taking this suggestion into account, I have revised the manuscript by elaborating on how the interview questions were developed. The revised version now includes a detailed explanation of the literature consulted and the expert feedback received during the development process. This additional information has been added under the section titled “Validity and Reliability of Research Data.”
Original paragraph:
With the aim of ensuring the research reliability, the participants were included in the research according to the principle of voluntariness and were asked to detail their in-dividual experiences in the process as much as possible. During the individual interviews, the researchers objectively listened to the participants. The researchers carried out the thematic analysis process independently of each other to increase the reliability of the research.
Response: In line with the reviewer’s suggestion, relevant explanations have been added to clarify the development process of the interview questions. Specifically, a detailed paragraph addressing this issue has been incorporated into the section titled “Validity and Reliability of Research Data,” on pages 5 and 6, between lines 229 and 253.
Added paragraph: Prior to the development of the interview questions, both the national and international literature were extensively reviewed. This process included an examination of measurement tools used in studies such as [22-19-28], as well as theoretical and thematic insights from research by [38-29-39]. The literature review guided the construction of the study’s theoretical framework and provided a comprehensive understanding of the research topic. Based on this framework, the interview questions were designed. To ensure the clarity, relevance, and content validity of the items, expert feedback was obtained from three field specialists. Necessary revisions were made in light of their evaluations, and the final version of the interview form was produced accordingly.
The semi-structured interview form that was prepared to explore the problems experienced during the subjects’ participation in search and rescue operations consisted of three main sections and 18 questions. In the first part, nine questions aimed to determine the sociodemographic characteristics of the participants and to identify whether they have previously intervened or witnessed traumatizing events (witnessing death, seeing severely injured, dismembered face, body, etc.) and whether they have received psychological support and first aid training before participating in search and rescue operations. In the second part, there were five questions about how they decided to participate in search and rescue operation; the region that they went to and the duration of their stay; whether they intervened or witnessed traumatizing events (witnessing death; severely injured, dismembered faces; seeing a body; etc.) during the operations; what the challenging conditions were; and the traumatic events that affected the participants most deeply. In the third and last section, four questions were asked to determine how the traumatic events that the participants experienced affected their inner world, whether they received support (professional, family, friends, etc.) after the operations, how the events affected their private relationships, and how they coped with the events.
Comment: 413- in the discussion section, you mention "the reasons behind the decisions" to engage, etc, but this is the first mention of this anywhere. Again, please define the questions asked in the study. It is qualitative, so the whole thing is the content.
Response: Thank you very much for your constructive and thoughtful comment. We really appreciate your observation regarding the phrase “reasons behind decisions” that was previously mentioned in the discussion section without sufficient contextual basis. In response to your suggestion, we have revised the previous sections of the manuscript to more clearly define both the research questions and the rationale behind the interview questions. In particular, a new paragraph has been added to the Methods section, under the subheading “Method”, “Data Collection Tools”, that outlines the structure and purpose of the interview guide, with special attention to how the questions were developed to explore the factors that influenced the participants’ decision-making processes. This new paragraph appears on page 6 of the revised manuscript. If the revisions still do not provide the necessary background for the wording in the discussion section, I will be happy to make further adjustments based on your suggestions.
Added Paragraph:
The semi-structured interview form that was prepared to explore the problems experienced during the subjects’ participation in search and rescue operations consisted of three main sections and 18 questions. In the first part, nine questions aimed to determine the sociodemographic characteristics of the participants and to identify whether they have previously intervened or witnessed traumatizing events (witnessing death, seeing severely injured, dismembered face, body, etc.) and whether they have received psychological support and first aid training before participating in search and rescue operations. In the second part, there were five questions about how they decided to participate in search and rescue operation; the region that they went to and the duration of their stay; whether they intervened or witnessed traumatizing events (witnessing death; severely injured, dismembered faces; seeing a body; etc.) during the operations; what the challenging conditions were; and the traumatic events that affected the participants most deeply. In the third and last section, four questions were asked to determine how the traumatic events that the participants experienced affected their inner world, whether they received support (professional, family, friends, etc.) after the operations, how the events affected their private relationships, and how they coped with the events.
In line with your previous and this comment, lines 229-259 on pages 5 and 6 of the article have been completely revised.
Comment: The beginning of the strengths and limitations section - if this paper is for an international audience, please define the event you are referring to, or exlude it? The litigation that is mentioned.
Response:
Thank you very much for your thoughtful and constructive comment. We appreciate your suggestion to clarify the reference to the litigation process at the beginning of the “Strengths and Limitations” section, especially for international readers.
We have therefore reorganized the text to briefly explain the relevant process. I would like to note that I am prepared to remove or further simplify this section if it is still considered redundant or too localized for the overall flow of the paper.
The relevant edit was made on page 2, line 71-85 of the study.
Revised and Added paragraph:
What distinguishes the TRNC’s participation from that of other countries is the particular nature of its connection to the affected regions. Many individuals from the TRNC were personally impacted, either through the loss of relatives or close emotional ties to those affected. Additionally, the deep-rooted historical, cultural, and political relationship between Turkey and Northern Cyprus further intensified the emotional engagement of TRNC citizens. As a result, this might have led to empathic identification among volunteers from the TRNC who participated in the rescue work.
A tragic event that brought this emotional impact to a peak was the collapse of the Grand Isias Hotel in Adıyaman. The building’s destruction resulted in the deaths of 35 TRNC citizens, including members of the GazimaÄŸusa Turkish Maarif College volleyball team who had traveled to Turkey to participate in a school sports competition [14]. This incident deeply affected the TRNC public, receiving extensive media coverage and generating a profound sense of collective grief [15]. In the days that followed, everyday life in the TRNC came to a halt; businesses closed, a period of national mourning was declared, and calls for justice for the victims became central to public discourse [16].
A few more thoughts - the selection of the particular responders could be more fleshed out, including the differences between these kind of responders, which you do get at in the questionnaire, by asking about training of the responders, but again, this is buried in the appendix. Those trained in rescue/response are more immune to post trauma symptoms than those not trained in it, based on literature - it is a protective factor. Some of these folks might have been trained in disaster response, perhaps the nurses? Likely not the volunteers?
Relevant additions are added on page 7 of the article between lines 285-295.
[16].Comment: For publication in English, the language must be improved. Perhaps a science interpreter could be employed for this purpose? Most of the errors are somewhat minor grammar or syntax errors, however, there are a number of errors that resulted in my not understanding the meaning of the text. It seems that an online translation service might have been used and unfortunately, there are times that, although the words might translate, the meaning does not.
Response:
We sincerely thank the reviewer for their careful reading and constructive feedback regarding the language and clarity of the manuscript. We fully acknowledge the importance of linguistic precision in ensuring the effective communication of scientific work.
The manuscript has undergone English language editing by MDPI. The text has been checked for grammar and correct use of common technical terms and edited to a level appropriate for reporting research in a scientific journal.
In response to the concerns raised, the revised version of the manuscript has been thoroughly edited using the journal’s professional English language editing service. This revision aimed to address all grammatical, syntactic, and stylistic issues that may have impeded comprehension in the initial submission.
Nevertheless, should any remaining ambiguities or unclear expressions be identified during further review, we remain fully open to implementing additional revisions in accordance with the reviewer’s recommendations.

Reviewer 2 Report
Comments and Suggestions for Authors
Unfortunately, limited number (eight) participants of this research constituted weak statistical its outcomes and suggests authors' to choice one of following options:
-
to withdraw the manuscript, make the same research including much more participants no less than 100
-
to change title of this manuscript to: ,, A Qualitative Preliminary Study on Secondary Trauma / Trauma Experiences of Individuals Participating in Search and Rescue Activities After Earthquake".
Author Response
Dear Reviewer (2),
Comment: Unfortunately, limited number (eight) participants of this research constituted weak statistical its outcomes and suggests authors' to choice one of following options: to withdraw the manuscript, make the same research including much more participants no less than 100 to change title of this manuscript to: ,, A Qualitative Preliminary Study on Secondary Trauma / Trauma Experiences of Individuals Participating in Search and Rescue Activities After Earthquake".
Response:
I appreciate your concern regarding the limited sample size of eight participants. However, I would like to emphasize that the small sample size aligns with the principles of qualitative research, where depth and detailed exploration are considered more important than generalizability. In qualitative research, it is widely accepted that a smaller, more focused sample can provide rich insights when exploring a specific phenomenon in depth (Patton, 2002; Creswell, 2013). The decision to work with a small group of participants was not intentionally made to increase the sample size, but rather to allow for a deeper examination of secondary trauma experiences within a specific group. This methodological approach differs from quantitative studies that aim for general conclusions based on larger sample sizes (Sandelowski, 2000; Mason, 2010).
Especially considering the qualitative and exploratory nature of our study, conducting it with a small sample is a methodologically used approach in the literature. In this context, there are similar studies conducted with a limited number of participants and published in reputable journals. For example: 41. Khatri J., Fitzgerald, G., & Chhetri, M. B. P. (2019). Health risks and challenges in earthquake responders in Nepal: a qualitative research. Prehospital and disaster medicine, 34(3), 274-281.
Original title: Secondary Trauma / Trauma Experiences of Individuals Par-ticipating in Search and Rescue Activities After Earthquake: A Qualitative Study
Revised title: A Qualitative Preliminary Study on Secondary Trauma Experiences of Individuals Participating in Search and Rescue Activities After an Earthquake
We would like to express our sincere thanks for your time and valuable contributions to the review process. In line with the suggestions provided by the reviewers, the manuscript has been carefully revised to improve its overall clarity, coherence, and academic depth.
If you have any additional suggestions that you believe would further enhance the manuscript, we would be more than happy to consider and incorporate them in the revision. We truly appreciate your guidance and support throughout this process.

Reviewer 3 Report
Comments and Suggestions for Authors
The authors present data from a qualitative analysis of professionals involved in rescue activities during an earthquake. The overall information provided in the manuscript is novel; as such, the content of the present manuscript has the potential to add to the current literature in an important new way.
Title: please avoid/; decide, whether you report secondary trauma or trauma experiences, but not both.
Abstract: please specify where this earthquake happened. Readers might not know where Kahramanmaras might be. Please add some more information as regards these aid volunteers; specifically, report their main age, and generation; likewise, report their functions during the search and rescue activities, including their professional background and their experiences in years of being a rescuer. Overall, the abstract is nicely written.
Introduction: after the first paragraph, please report references. References are timely; the introduction provides an nicely crafted and thoroughly elaborated overview of possible psychological issues in the aftermath of disasters such as earthquakes among professional workers. Importantly, the authors of the present manuscript also summarized the current state of the art as regards post traumatic growth.
Consider to put the aims of the present study into past tense, given that you already did complete the study. As such, the word should be: “this study aimed to examine”.
Methods: please describe with more details the inclusion and exclusion criteria; likewise, describe in more details how and why these eight people volunteered to participate to this current study.
Result section was particularly nicely crafted. One subtitle should be “seeing that children”.
Discussion: also this part of the manuscript was nicely crafted.
Author Response
Dear Reviewer (3),
Comment: The authors present data from a qualitative analysis of professionals involved in rescue activities during an earthquake. The overall information provided in the manuscript is novel; as such, the content of the present manuscript has the potential to add to the current literature in an important new way.
Response: Thank you for your positive feedback. We greatly appreciate your recognition of the novelty and potential contribution of our work to the existing literature. We are committed to improving the quality of our manuscript and are fully prepared to make any necessary revisions. Should you have any specific suggestions or areas requiring further clarification, we are happy to address them.
Comment: Title: please avoid/; decide, whether you report secondary trauma or trauma experiences, but not
Response: Thank you for your comment and suggestion regarding the title. We have revised the title to better reflect the scope and content of the study, taking into account feedback from both reviewers, as well as criticisms regarding clarity and focus.
Original Title: "Secondary Trauma / Trauma Experiences of Individuals Participating in Search and Rescue Activities After Earthquake: A Qualitative Study"
Revised Title: " A Qualitative Preliminary Study on Secondary Trauma Experiences of Individuals Participating in Search and Rescue Activities After an Earthquake"
Comment: Please specify where this earthquake happened. Readers might not know where Kahramanmaras might be. Please add some more information as regards these aid volunteers; specifically, report their main age, and generation; likewise, report their functions during the search and rescue activities, including their professional background and their experiences in years of being a rescuer.
Response: Thank you for your thoughtful and constructive feedback on the abstract. In response to your suggestion, we have revised the abstract to include additional context regarding the location of the earthquake, specifying that it occurred in the southeastern region of Türkiye, with KahramanmaraÅŸ as the epicenter. We have also added demographic information about the aid volunteers, including their age range and generational group, along with a brief description of their roles during the search and rescue process, their professional backgrounds, and their level of prior experience in disaster response. We hope these additions enhance the clarity and comprehensiveness of the abstract.
Original abstract:
Abstract: Background:This study aims to analyze the challenges of professionals and volunteers in search and rescue operations during the KahramanmaraÅŸ earthquake dated 6th February 2023. Method: This study presents the results of semi-structured in-dividual interviews with eight purposively sampled volunteers who participated in the search and rescue activities. The research data were analyzed with MAXQDA Analytic Pro 2020 software. Results: Within the scope of this research, four main themes and 21 sub-themes were identified. The first theme is related to the nature of the traumatic events and reflects the characteristics of the traumatic experiences of the par-ticipants. The second theme is secondary trauma symptoms and shows that the participants experienced symptoms such as overstimulation, intrusive thoughts, sleep problems, anger and concentration difficulties. The third theme focuses on posttraumatic growth symptoms. Participants reported experiencing developmental changes following trauma, such as changes in self-perception, the ability to recognize new situations, understanding the value of life, and positive relationships related to personal growth. Finally, the fourth theme is related to coping skills used to cope with traumatic events; participants shared their coping strategies and the impact of these strategies. Conclusion: The study highlights the necessity for an assessment of individuals in search and rescue operations in terms of secondary trauma. The study findings may be used as a reference to develop post-disaster psychosocial support services for volun-teer search and rescue teams. Additionally, research findings can be used to renew the content of pre-field preparation trainings.
Revised abstract:
Abstract: Background: This study aimed to analyze the challenges faced by professionals and volunteers in search and rescue operations after the earthquake that struck the southeastern region of Turkey, with its epicenter in KahramanmaraÅŸ, on February 6, 2023. Method: This research was conducted using a qualitative approach; specifically, a phenomenological method. It presents the results of semi-structured individual interviews with eight sampled volunteers who participated in the search and rescue activities following the earthquake. Participants were between the ages of 24 and 45, of which 3 were nurses, 3 were journalists, and 2 were civilian volunteers with no formal training in search and rescue. In terms of nationality, five participants were citizens of the Turkish Republic of Northern Cyprus (TRNC) only, while three had both TRNC and Turkish citizenship. Field duties included providing medical support, documenting incidents, and assisting survivors in collaboration with civil society organizations. The research data were analyzed using MAXQDA Analytic Pro 2020. Results: Within the scope of this research, four main themes and twenty-one sub-themes were identified. The first theme is related to the nature of the traumatic events and reflects the characteristics of the traumatic experiences of the participants. The second theme is secondary trauma symptoms, showing that the participants experienced symptoms such as overstimulation, intrusive thoughts, sleep problems, anger, and concentration difficulties. The third theme focuses on post-traumatic growth symptoms. Participants reported experiencing developmental changes following trauma, such as changes in self-perception, the ability to recognize new situations, understanding the value of life, and positive relationships related to personal growth. Finally, the fourth theme is related to coping skills used to cope with traumatic events; participants shared their coping strategies and the impact of these strategies. Conclusion: This study highlights the need to assess individuals in search and rescue operations in terms of secondary trauma. Our findings may be used as a reference to develop post-disaster psychosocial support services for volunteer search and rescue teams. Additionally, the findings can be used to renew the content of pre-field preparation trainings.
Comment: Introduction: after the first paragraph, please report references. References are timely; the introduction provides an nicely crafted and thoroughly elaborated overview of possible psychological issues in the aftermath of disasters such as earthquakes among professional workers. Importantly, the authors of the present manuscript also summarized the current state of the art as regards post traumatic growth.
Response: Thank you for your constructive evaluation of the introduction. In line with your comment, the introduction section on the psychological effects on professionals after disasters and post-traumatic growth has been revised and current studies in this field have been added to the text. With this revision, we aimed to establish a more solid and up-to-date foundation for the theoretical framework of our study. We would like to state that your feedback has contributed to the development of the academic quality of our article. Newly added studies can be found on page two of the revised manuscript, lines 44 to 57.
Comment: Consider to put the aims of the present study into past tense, given that you already did complete the study. As such, the word should be: “this study aimed to examine”.
Response: Thank you for this thoughtful suggestion. We agree with the reviewer’s observation. As the study has already been completed, we revised the aim statement accordingly and changed the verb tense to past tense.
Original sentence: “This study aims to examine the secondary trauma experiences…”
Revised sentence: “This study aimed to examine the secondary trauma experiences…”
In addition, a revised version of the manuscript was additionally English language edited by MDPI. The text has been checked for grammar and correct use of common technical terms and edited to a level appropriate for reporting research in a scientific journal.
Comment: Methods: please describe with more details the inclusion and exclusion criteria; likewise, describe in more details how and why these eight people volunteered to participate to this current study.
Response: Thank you for this valuable comment. We have revised the methodology section to include more detailed information about the inclusion and exclusion criteria. Specifically, individuals were included in the study if they (1) participated voluntarily in search and rescue activities following the February 6, 2023 earthquake in Türkiye, (2) were over 18 years of age, and (3) agreed to participate in the study by providing informed consent. Individuals who did not directly participate in the field or who withdrew consent were excluded.
Furthermore, we added more detail about how and why these eight individuals volunteered to participate in the study. Due to legal and institutional access restrictions, participants were recruited using the snowball sampling method. Those who initially participated referred others who had similar experiences.
Finally, in response to your request for more detailed information about the participants in this section of the abstract, we have added comprehensive details about the participants to the Methodology section of the article. These details include demographic characteristics, roles, and additional information about the responsibilities they assumed during field operations.
Revised paragraph: As institutional participation in this research was not approved, only individuals who volunteered for search and rescue work with their own resources could be included in the study. Eight participants were reached using the snow-ball technique. This method is particularly effective for reaching hidden or difficult-to-access populations, as it enables participant recruitment through a chain of referrals [35]. The participants arrived at the disaster area between February 6th and 9th and remained active in the field for periods ranging from 4 to 10 days. Nurses took on tasks such as providing basic medical services, administering first aid, and assisting in the care of the injured. Journalists were not only responsible for documenting the events and informing the public but also supported the processes of body bagging and victim identification, which involved emotionally and physically demanding tasks. Civil volunteers contributed to rubble removal, excavation support, and various humanitarian aid activities.
Newly added studies can be found on page 4 of the revised manuscript, lines 154 to 165.
Comment: Result section was particularly nicely crafted. One subtitle should be “seeing that children”.
Response: Thank you for your positive feedback on the Results section. We greatly appreciate your acknowledgment of the structure and clarity of this part of the manuscript
You can find the change on page 9, line 351 of the article.
Comment: Discussion: Also this part of the manuscript was nicely crafted.
Response: Thank you for your positive feedback on the Discussion section.
We would like to express our sincere thanks for your time and valuable contributions to the review process. In line with the suggestions provided by the reviewers, the manuscript has been carefully revised to improve its overall clarity, coherence, and academic depth.

Reviewer 4 Report
Comments and Suggestions for Authors
Thank you for allowing me to review the article titled: “Secondary Trauma / Trauma Experiences of Individuals Participating in Search and Rescue Activities After an Earthquake: A Qualitative Study”
Abstract:
Include the study's objective on secondary trauma in rescue volunteers, mention the phenomenological methodology used, highlight key outcomes such as coping strategies and posttraumatic growth, and briefly outline practical implications for implementing emotional support programs.
Introduction:
Clarify the relationship between secondary trauma and rescue operations, emphasizing how this study contributes to current knowledge in this area.
Include previous studies on secondary trauma in rescue volunteers and highlight the knowledge gap.
Further develop the section on secondary trauma.
Expand on the conditions and magnitude of the KahramanmaraÅŸ earthquake and its relationship to the experiences of rescue volunteers.
Materials and Methods:
Explain why the phenomenological approach is appropriate for studying secondary trauma in rescue volunteers.
Describe how safe psychological conditions were ensured during the interviews, explaining the comfortable and private environment and the process, as well as the immediate emotional support available to participants when addressing sensitive topics.
Break the long paragraph in section 2.5, "Validity and Reliability of Research Data," into clearer and more structured sentences to improve readability and comprehension.
Results:
Adjust the indentation of the title "Symptoms of Secondary Trauma" in Table 3, as it appears shifted to the right.
If the interviews were conducted in Turkish, explain how the accuracy and fidelity of the translation and presentation of the results in English were ensured.
Discussion:
Explore how the study’s findings align with or contrast with previous studies and include more recent references that address both secondary trauma and posttraumatic growth.
Include practical recommendations for training and emotional support programs for rescue volunteers to mitigate the effects of secondary trauma.
Author Response
Dear Reviewer (4),
Comment: Thank you for allowing me to review the article titled: “Secondary Trauma / Trauma Experiences of Individuals Participating in Search and Rescue Activities After an Earthquake: A Qualitative Study”
Response: We truly appreciate the time and effort you dedicated to reviewing the paper.
Comment:
Abstract: Include the study's objective on secondary trauma in rescue volunteers, mention the phenomenological methodology used, highlight key outcomes such as coping strategies and posttraumatic growth, and briefly outline practical implications for implementing emotional support programs.
Response: Thank you for your constructive feedback and valuable suggestions. In response to your comment regarding the study description, we have revised the paragraph to improve clarity and flow.
Original Abstract: Abstract: Background:This study aims to analyze the challenges of professionals and volunteers in search and rescue operations during the KahramanmaraÅŸ earthquake dated 6th February 2023. Method: This study presents the results of semi-structured in-dividual interviews with eight purposively sampled volunteers who participated in the search and rescue activities. The research data were analyzed with MAXQDA An-alytic Pro 2020 software. Results: Within the scope of this research, four main themes and 21 sub-themes were identified. The first theme is related to the nature of the trau-matic events and reflects the characteristics of the traumatic experiences of the par-ticipants. The second theme is secondary trauma symptoms and shows that the par-ticipants experienced symptoms such as overstimulation, intrusive thoughts, sleep problems, anger and concentration difficulties. The third theme focuses on posttrau-matic growth symptoms. Participants reported experiencing developmental changes following trauma, such as changes in self-perception, the ability to recognize new situ-ations, understanding the value of life, and positive relationships related to personal growth. Finally, the fourth theme is related to coping skills used to cope with traumatic events; participants shared their coping strategies and the impact of these strategies. Conclusion: The study highlights the necessity for an assessment of individuals in search and rescue operations in terms of secondary trauma. The study findings may be used as a reference to develop post-disaster psychosocial support services for volun-teer search and rescue teams. Additionally, research findings can be used to renew the content of prefield preparation trainings.
Revised Abstract: Background: This study aimed to analyze the challenges faced by professionals and volunteers in search and rescue operations after the earthquake that struck the southeastern region of Turkey, with its epicenter in KahramanmaraÅŸ, on February 6, 2023. Method: This research was conducted using a qualitative approach; specifically, a phenomenological method. It presents the results of semi-structured individual interviews with eight sampled volunteers who participated in the search and rescue activities following the earthquake. Participants were between the ages of 24 and 45, of which 3 were nurses, 3 were journalists, and 2 were civilian volunteers with no formal training in search and rescue. In terms of nationality, five participants were citizens of the Turkish Republic of Northern Cyprus (TRNC) only, while three had both TRNC and Turkish citizenship. Field duties included providing medical support, documenting incidents, and assisting survivors in collaboration with civil society organizations. The research data were analyzed using MAXQDA Analytic Pro 2020. Results: Within the scope of this research, four main themes and twenty-one sub-themes were identified. The first theme is related to the nature of the traumatic events and reflects the characteristics of the traumatic experiences of the participants. The second theme is secondary trauma symptoms, showing that the participants experienced symptoms such as overstimulation, intrusive thoughts, sleep problems, anger, and concentration difficulties. The third theme focuses on post-traumatic growth symptoms. Participants reported experiencing developmental changes following trauma, such as changes in self-perception, the ability to recognize new situations, understanding the value of life, and positive relationships related to personal growth. Finally, the fourth theme is related to coping skills used to cope with traumatic events; participants shared their coping strategies and the impact of these strategies. Conclusion: This study highlights the need to assess individuals in search and rescue operations in terms of secondary trauma. Our findings may be used as a reference to develop post-disaster psychosocial support services for volunteer search and rescue teams. Additionally, the findings can be used to renew the content of pre-field preparation trainings.
Changes to the abstract can be found on pages 1, lines 12-38.
Comment:
Introduction: Clarify the relationship between secondary trauma and rescue operations, emphasizing how this study contributes to current knowledge in this area.
Response: Thank you for your valuable comment. Thank you for your suggestion to clarify the relationship between secondary trauma and rescue operations.
In response to this, we have revised the introduction and discussion section in particular to clearly highlight how the experiences of individuals involved in search and rescue operations can lead to secondary trauma, particularly through exposure to distressing events such as death and witnessing suffering.
Changes to the introduction can be found on pages 2 and 3, lines 88-114.
Changes to the discussion section can be found on page 16 in lines 626-645.
Comment: Introduction: Include previous studies on secondary trauma in rescue volunteers and highlight the knowledge gap.
Response: Thank you for your insightful comment. In response, we have revised the introduction to include a review of previous studies on secondary trauma in rescue volunteers. We have highlighted the existing research that addresses the psychological impact of participating in disaster response efforts, focusing on the experiences of both professionals and volunteers.
Comment: Introduction: Further develop the section on secondary trauma.
Response: Thank you for your suggestion. In response, we have further developed the secondary trauma section, examining its theoretical basis and its relationship to rescue operations in more detail. You can see the changes on 3 pages between lines 93-114.
Comment: Expand on the conditions and magnitude of the KahramanmaraÅŸ earthquake and its relationship to the experiences of rescue volunteers.
Thank you for your valuable feedback. In line with your suggestion, we have taken a more detailed look at the circumstances surrounding the KahramanmaraÅŸ earthquake and its magnitude, as well as its relationship with the experiences of search and rescue volunteers. In our study, we emphasized how the magnitude of the earthquake and the traumatic scenes that rescue volunteers witnessed - collapsed buildings, large-scale loss of life and emotional distress caused by the loss of children or loved ones - affected the volunteers. The introduction of the study has been completely revised.
Comment: Materials and Methods: Explain why the phenomenological approach is appropriate for studying secondary trauma in rescue volunteers.
Response: Your comments and feedback have contributed significantly to the deepening and more solid foundation of our work. I would like to state that the phenomenological approach is discussed in detail in the article. If further clarification is needed, I would like to state that we are ready to make the necessary corrections and provide more information. Changes to the method section can be found on page 4, lines 141-151.
Revised sentence:
This study was conducted using a qualitative method, with qualitative data col-lection methods such as observation, interviews, or document analysis utilized accordingly [33]. We used a qualitative research method known as a “phenomenological study” [34]. The phenomenological approach seeks to deeply understand individuals' experiences, and this study specifically examines the challenges and traumatic effects faced by those involved in search and rescue activities. Through focusing on the differential impact of trauma and the emotional and psychological meaning-making of participants, this approach offers comprehensive insights into how secondary trauma is experienced.
Comment: Materials and Methods: Describe how safe psychological conditions were ensured during the interviews, explaining the comfortable and private environment and the process, as well as the immediate emotional support available to participants when addressing sensitive topics.
Response: Thank you for your valuable feedback. In response, we have elaborated on the procedures undertaken to ensure the psychological safety of participants during the interviews.
Original sentence: Each interview lasted approximately fifty minutes and was conducted in the therapy room in the first author's clinic. Data saturation was reached in this process.
Revized sentence:
All interviews were conducted by the first author, a licensed clinical psychologist and certified positive psychotherapist. In addition to verbal content, the researcher took brief notes on participants’ emotional reactions and non-verbal cues throughout the sessions. When sensitive topics arose, the researcher maintained an empathic and supportive stance, pacing the conversation in accordance with the participant’s emotional state. If participants experienced emotional discomfort during the interviews, grounding or breathing techniques were applied as needed. Furthermore, participants were provided with information on how to access further psychological support if they desired. The researcher had access to a professional referral network, consisting of trusted therapists from various fields, and participants assessed as possibly needing further support were referred accordingly. No urgent psychological intervention was required during or following any of the interviews.
Changes to the method section can be found on page 5, lines 198-209.
Comment: Materials and Methods:
Break the long paragraph in section 2.5, "Validity and Reliability of Research Data," into clearer and more structured sentences to improve readability and comprehension.
Response: Thank you for your valuable feedback. In line with your suggestion, we have revised paragraph 2.5, “Validity and Reliability of Research Data” to improve clarity and readability.
The relevant section has been revised for clarity to avoid ambiguity, and the academic language of the article has been reorganized using the MDPI English translation service.
Comment: Results: Adjust the indentation of the title "Symptoms of Secondary Trauma" in Table 3, as it appears shifted to the right.
Response: Thank you for your valuable observation. In response to your comment, we have adjusted the indentation of the title "Symptoms of Secondary Trauma" in Table 3, as it appeared shifted to the right. The title has now been properly aligned to ensure consistency and clarity throughout the manuscript.
Comment: Results: If the interviews were conducted in Turkish, explain how the accuracy and fidelity of the translation and presentation of the results in English were ensured.
If the interviews were conducted in Turkish, explain how the accuracy and fidelity of the translation and presentation of the results in English were ensured.
Response: Thank you for your comment. In order to ensure the reliability of the data in this study, certain steps were followed as stated in the article. First of all, in line with their professional experiences, theoretical knowledge, academic study experiences and observations, the researchers conducted their research to determine the physical, emotional and social effects that participation in search and rescue activities can have on individuals' lives and the effects of these effects on posttraumatic growth.
In order to ensure the reliability of the research, participants were included in the study on a voluntary basis. Throughout the process, participants were asked to share their individual experiences in as much detail as possible. During the interviews, the researcher listened to the participants impartially to allow them to freely express their experiences, emotional experiences and thoughts, and did not confirm or deny them.
Finally, the interviews were conducted in Turkish and the data were transcribed verbatim from the audio recordings in Turkish and then translated into English by a researcher who has an advanced command of both Turkish and English and is experienced in research translation. To ensure the quality and accuracy of the translation, the translated text was reviewed and approved by a second bilingual researcher.
Comment: Discussion: Explore how the study’s findings align with or contrast with previous studies and include more recent references that address both secondary trauma and posttraumatic growth.
Response: Thank you for your valuable suggestion. In response, the discussion section has been revised to include a deeper comparison between the current findings and recent studies with individuals exposed to the same earthquake. Recent and relevant literature on both secondary traumatic stress and posttraumatic growth has been integrated to strengthen the contextual basis of the results. These additions aim to better reflect how the current study fits with and contributes to the evolving body of knowledge in this field. The revised chapter can be found on page 16 of the manuscript. If deemed necessary by the referees or the editorial board, we are open to making further additions or revisions to improve the clarity and scientific contribution of the article.
Changes to the discussion section can be found on page 16, lines 625-645.
We would like to express our sincere thanks for your time and valuable contributions to the review process. In line with the suggestions provided by the reviewers, the manuscript has been carefully revised to improve its overall clarity, coherence, and academic depth.
